# Evaluation of the Associative Effects of Rice Straw with Timothy Hay and Corn Grain Using an In Vitro Ruminal Gas Production Technique

**DOI:** 10.3390/ani10020325

**Published:** 2020-02-18

**Authors:** Ling Sun, Mingyung Lee, Seoyoung Jeon, Seongwon Seo

**Affiliations:** Division of Animal and Dairy Sciences, Chungnam National University, Daejeon 305-764, Korea; sunlingcnu@gmail.com (L.S.); mingyung1203@cnu.kr (M.L.); seoyoung203@cnu.kr (S.J.)

**Keywords:** associative effects, in vitro automated gas production system, rice straw

## Abstract

**Simple Summary:**

Rice straw is a widely used forage source for ruminants in most Asian countries; thus, it is important to accurately estimate its nutritional value. Rice straw is typically fed to the animals along with other ingredients, and the associative effects of the combined ingredients may alter the nutritional value of rice straw. We found associative effects on the ruminal fermentability (gas production kinetics and rumen parameters), especially when rice straw was co-fermented with timothy hay and corn grain. We conclude that the nutritional value of rice straw increases when used with timothy hay and corn grain, due to the associative effects among feeds, which should be considered in diet formulations.

**Abstract:**

The objective of this study was to evaluate the associative effects of rice straw with timothy hay and corn grain. Using an automated gas production system, in vitro ruminal fermentation was studied for six substrates: 100% rice straw, 100% timothy hay, 100% corn grain, 50% rice straw and 50% timothy hay, 50% rice straw and 50% corn grain, and 50% rice straw, 25% timothy hay, and 25% corn grain. Incubation was performed in three batches with different rumen fluids to assess the in vitro ruminal gas production kinetics and rumen parameters (pH, NH_3_-N, volatile fatty acid (VFA), and true dry matter digestibility (TDMD)). The associated effects were tested by comparing the observed values of the composited feeds and the weighted means of individual feeds. There was a significant increase in NH_3_-N when rice straw was fermented with timothy hay, corn grain, or both (*p* < 0.05). TDMD increased when corn grain was co-fermented, and the total gas and VFA production increased when all three feeds were co-fermented. We conclude that the feed value of rice straw increases when fed to animals along with timothy hay and corn grain.

## 1. Introduction

Ration formulation is a mixture of individual feed ingredients, and the metabolizable energy (ME) and net energy (NE) of the ration are assumed to be the sum of the individual ingredients. This assumes that the ME and NE values of the individual ingredients do not change when mixed with other feed products. However, some studies have documented associative effects among feed ingredients [1,2,3], defined as interactions between the ration components that alter the nutritional value of the individual ingredients [4]. Associative effects may be positive, negative, or absent [5], and though widely discussed in theory, they are seldom considered in feed formulation.

Ruminant animals are fed diets of highly fibrous roughage, grains, brans, and hulls. Associative effects may be an important factor in ruminant rations, as interactions among these ingredients could modify the microbial fermentation processes in the rumen [6,7]. Several studies have investigated the associative effects of barley straw and alfalfa (*Medicago sativa*) [8], grasses and legumes [9], red clover (*Trifolium pratense*) and kikuyu grass silage [10], and corn stalks and alfalfa (*Medicago sativa*) [11].

Rice is a major crop in Asia and, consequently, rice straw is a common roughage for ruminant animals in most Asian countries. In 2014, approximately 538.88 million tons of rice straw, accounting for more than 90% of total world production, was produced in Asia [12]. Prior studies have investigated the associative effects of rice straw with alfalfa hay and corn silage [13], and those of chemically treated rice straw with grass hay or mulberry leaves [6]. To date, however, no studies have tested the associative effects of rice straw with timothy hay and corn grain, which are commonly co-fed ruminants such as beef cattle. In particular, timothy hay is well known for its good quality and contributes a major proportion of imported forage in Korea [14]. Compared to rice straw, timothy hay contains easily digestible carbohydrates, so that it has higher digestibility of fiber and dry matter [15]. Timothy hay, however, is two to three times more expensive than rice straw, and thus it is mainly used as a forage source for calves and lactating dairy cows and as a supplement for rice straw for growing cattle in Korea [16].

In vitro fermentation methods are widely used to study associative effects, as the digestibility and rumen fermentation of feed can be measured in a relatively simple manner, and numerous samples can be evaluated at one time [17]. Among different in vitro methods, the in vitro gas production technique is commonly used to assess the differences between single and mixed feed substrates for different variables [6,7,13,18,19]. An automated in vitro gas production system (AGPS) was first developed by Pell and Schofield [20]—the AGPS continuously measures the gas produced during in vitro ruminal fermentation with electronic pressure transducers, which are sensitive enough to detect small changes due to associative effects. The AGPS developed by Pell and Schofield is inexpensive, easily adopted, and simple to maintain.

The objective of this study was to evaluate the associative effects of rice straw, timothy hay, and corn grain on the ruminal fermentation characteristics using an automated in vitro gas production system.

## 2. Materials and Methods

The cannulated cattle used in this study were maintained at the Center for Animal Science Research, Chungnam National University, Korea. All animal usage and experimental procedures were conducted with the approval of the Chungnam National University Animal Research Ethics Committee (CNU-00830).

### 2.1. Preparation of Experimental Diets

To measure the in vitro rumen fermentation characteristics, the feedstuffs were incubated separately or in combination as follows: 100% rice straw (R), 100% timothy hay (T), 100% corn grain (C), 50% rice straw and 50% timothy hay (RT), 50% rice straw and 50% corn grain (RC), and 50% rice straw, 25% timothy hay, and 25% corn grain (RTC). The chemical compositions of rice straw, timothy hay, and corn grain are presented in Table 1. The feed samples were ground in a cyclone mill (Foss Tecator Cyclotec 1093, Foss, Hillerød, Denmark) with 1 mm sieves prior to the chemical analyses and measurements of in vitro gas production.

### 2.2. Automated Gas Production System

For this study, we developed and used a computerized in vitro gas production system, similar to Pell and Schofield [20]. The system consisted of an incubator (DS-110S, Daewon science, Bucheon-si, Korea), two 10 multi-plate stirrers (MS-52M, Jeio-Tech, Seoul, Korea), pressure sensors attached to the incubation bottles (four arrays of five 125 mL serum bottles, for a total 20 bottles), and a 20-channel data logger. The dimensions of the incubator were 600 × 580 × 650 (W × D × H, unit: mm), the internal temperature was maintained at 39 °C, and a pressure sensor (WIKA A-10; Wika, Lawrenceville, GA, USA) measured pressure increases up to 15 psi (pound-force per square inch; i.e., 1 standard atmosphere, atm). The data logger converted the analog signals from the pressure sensors into digital signals and stored them, and the data were transferred to a computer (via a USB drive) for processing using Microsoft Excel Macros.

In the in vitro fermentation study, the gas produced during anaerobic fermentation accumulated in the bottles. A leakage test and sensor calibration were performed prior to each run of the system, and the contents of the fermentation bottles were stirred continuously with a magnetic bar (90 rpm). The internal pressure of the bottles was measured by pressure sensors, and the data logger automatically recorded the values every 20 min.

### 2.3. In Vitro Ruminal Fermentation

Two cannulated, non-lactating Holstein cows were used as donor animals. Both were fed the same diet ad libitum twice daily (08:00 and 18:00): 600 g/kg corn silage (232 g/kg DM on as-fed basis; 121 g/kg CP, 527g/kg neutral detergent fiber (NDF) analyzed with a heat-stable amylase and expressed inclusive of residual ash (aNDF), 338 g/kg ADF, 27 g/kg EE, and 78 g/kg ash on DM basis) and 400 g/kg commercial concentrate (878 g/kg DM on as-fed basis; 171 g/kg CP, 339 g/kg aNDF, 192 g/kg ADF, 35 g/kg EE, and 59 g/kg ash on DM basis). The rumen fluid from the two donors was mixed and transferred to a Duran bottle on ice, measured for pH with a general pH meter (EcoMet P25, Istek, Inc., Seoul, Korea), and transported to the laboratory. The rumen contents were strained through eight layers of cheesecloth and glass wool and then combined with four times the volume of the in vitro solution described by Goering and Van Soest [21] under strictly anaerobic conditions. Twenty milliliters of the rumen fluid and buffer mixture was transferred to 125 mL serum bottles (Wheaton, Millville, NJ, USA) containing 0.2 g of the feed samples under continuous O_2_-free CO_2_ flow to maintain anaerobic conditions. The 125 mL serum bottles were sealed with septum stoppers and aluminum caps and connected to a three-way stopcock and 20 G needle set. The serum bottles were then connected to the sensor and incubated for 48 h at 39 °C. The contents of the fermentation bottles were stirred continuously with a magnetic bar at 90 rpm. In each fermentation batch, the incubation samples were performed in triplicate, and two blanks contained the buffered rumen fluid without the feed substrate.

### 2.4. Measurements and Chemical Analysis

After 48 h of incubation, the pH was measured and then volatile fatty acid (VFA) analysis was performed following the protocol proposed by Erwin et al. [22], using a gas chromatograph (HP 6890, Hewlett-Packard CO., Palo Alto, CA, USA) equipped with a flame ionization detector and capillary column (Nukol fused silica capillary column 30 m × 0.25 mm × 0.25 μm, Supelco, Inc., Bellefonte, PA, USA). The temperatures of the oven, injector, and detector were 90–180, 185, and 210 °C, respectively. Nitrogen was used as the carrier gas at a flow rate of 40 mL/min. NH_3_-N was analyzed as in Chaney and Marbach [23], where the NH_3_-N concentration was determined by measuring the absorbance at 630 nm with a spectrophotometer (UV-1800, Shimadzu Inc., Kyoto, Japan). The remaining undigested samples and fluid were used to measure aNDF to determine the NDF digestibility and true dry matter digestibility (TDMD) using a modified version of the methods proposed by Goering and Van Soest [21].

### 2.5. Calculation of Associative Effects

Associative effects were calculated using the following the method, which was similar to the one described by Liu [6]. The associative effects were tested by comparing the observed and predicted values of the composited feeds used in this study. The observed values were from the fermentation of the composited feeds (i.e., RT, RC, and RTC), and the predicted values were the weighted mean of the values for the fermentation of the individual feeds (i.e., R, T, and C). For example, the predicted value of RT = (the observed values for R multiplied by the proportion of R in RT; i.e., 0.5) + (the observed values for T multiplied by the proportion of T in RT; i.e., 0.5).

### 2.6. Statistical Analysis

The dual-pool Logistic equation [24] was fit to the gas production data using the NLIN (nonlinear regression) procedure in SAS (SAS Institute Inc., Cary, NC, USA) as follows: $$V_{t}=\frac{V_{1max}}{1+exp\left\{2+4\times \frac{k_{1}}{V_{1max}}\times \left(L-t\right)\right\}}+\frac{V_{2max}}{1+exp\left\{2+4\times \frac{k_{2}}{V_{2max}}\times \left(L-t\right)\right\}}$$
where *V_t_* is the total gas production at time t, *exp* is the exponential function, *V*_1*max*_ is the maximum gas production of the fast pool (mL), *k*_1_ is the maximum rate of gas production of the fast pool (h^−1^), *L* is the discrete lag time (h), *V*_2*max*_ is the maximum gas production of the slow pool (mL), and *k*_2_ is the maximum rate of gas production of the slow pool (h^−1^).

All data were analyzed using the MIXED procedure in SAS version 9.4 (SAS Institute Inc., Cary, NC, USA). Differences among the treatments were compared using Tukey’s range test, with a statistical significance threshold at *p* < 0.05 and a trending threshold at 0.05 ≤ *p* < 0.10. A paired t-test was used to compare the associative effects.

## 3. Results

### 3.1. Rumen Fermentation Characteristics and the Kinetics of Gas Production

The in vitro fermentation characteristics after 48 h incubation are presented in Table 2. There was a significant difference in pH, NH_3_-N, TDMD, propionate, and butyrate among all of the treatments (*p* < 0.05). The pH was lowest in the corn grain (*p* < 0.01) and higher in the rice straw than in the three mixed feed treatments (RT, RC, and RTC; Table 2). Compared to the rice straw and corn grain treatments, the concentration of NH_3_-N was significantly higher in all of the timothy hay-containing treatments (T, RT, and RTC). The TDMD was highest for corn grain and higher for timothy hay than for rice straw (*p* < 0.05). Moreover, the TDMD was significantly higher in the mixed feed treatments than in the rice straw treatment (*p* < 0.05). The butyrate concentration was also significantly higher in the corn mixed treatments than in the rice straw treatment (*p* < 0.05). However, there were no differences in the acetate and propionate concentrations of rice straw and the mixed feed treatments.

The kinetics of gas production are shown in Table 2. *V*_1*max*_ was greater for corn grain than for the other treatments (*p* < 0.05), and it was significantly increased in the corn mixed treatments compared to rice straw (*p* < 0.01). There were no significant differences in the discrete lag time of the rice straw and the mixed treatments. *V_max_* (the asymptotic total gas production, *V*_1*max*_ + *V*_2*max*_) was highest in the corn grain treatment but did not differ between the mixed and rice straw treatments.

The automated gas production profiles of the individual feedstuffs (rice straw, timothy hay, and corn grain) and the mixed feed treatments (RT, RC, and RTC) are shown in Figure 1. The cumulative volume of gas increased with more incubation time, and after 48 h of fermentation, the gas production was highest in the corn grain, followed by RC, RTC, timothy hay, RT, and rice straw.

### 3.2. Associative Effects on the Rumen Fermentation Characteristics

The probabilities of the significance of associative effects are shown in Table 3. There was a significant difference in the observed and predicted values of NH_3_-N and TDMD (except for RT) for all three mixed treatments (*p* < 0.05 for all), indicating a positive associative effect. The observed value of total VFA was significantly higher than the predicted value in the RTC treatment. There were no significant differences in the observed and predicted values of pH, individual VFA, or the acetate to propionate (A/P) ratio for the three mixed treatments. In the RTC treatment, the observed value of the discrete lag time was significantly decreased (*p* < 0.05), and the observed value of *V*_1*max*_ was significantly increased (*p* < 0.05) compared to the predicted values.

## 4. Discussion

Rice straw is a commonly used forage in East Asia, therefore, we were interested in the associative effects of other feeds with rice straw. To date, no studies have investigated the associative effects of rice straw, timothy hay, and corn grain. Corn is an important grain source for cattle in Korea, so there is great interest in the associative effects of corn grain and rice straw. Furthermore, the associative effects of timothy hay, a high-quality forage source, may compensate for the lack of nutrients in rice straw. The objective of this study was to determine the associative effects of rice straw with timothy hay and corn grain using an automated gas production system in an in vitro fermentation experiment.

In this study, the pH was significantly lower for the three mixed feed treatments (RT, RC, and RTC) than for the rice straw (*p* < 0.05), corresponding to higher total VFA concentrations in the mixed treatments (which contained more fermentable carbohydrates than the rice straw). The non-fiber carbohydrate (NFC) content of rice straw, timothy hay, and corn grain were 86, 204, and 792 g/kg DM, respectively (Table 1). NFCs provide ruminants with fermentable carbohydrates such as starch, sugar, and pectin, and promote the growth of rumen microorganisms. It has been shown that high dietary NFC increases the total VFA production, decreases ruminal pH, and modifies the molar proportions of VFAs by decreasing the ratio of acetic acid/propionate acid and increasing the proportion of butyrate [25]. Our results are consistent with Haddad [8], who supplemented barley straw diets with 300 and 450 g alfalfa hay (i.e., more NFC) and observed higher total VFA and lower pH values than in non-supplemented barley straw.

The NH_3_-N concentration, TDMD, and *V*_1*max*_ (except for RT) were significantly higher in the mixed feed treatments (RT, RC, and RTC) than in the rice straw treatment (*p* < 0.05). This can be explained by the higher NFC and lower NDF contents in the mixed feed treatments, which provided more fermentable carbohydrates to the rumen microorganisms [26]. Previous studies have also reported a positive correlation between gas and VFA production, and total gas production at 48 h was primarily determined by the NFC content [27,28]. The ash content of rice straw in this study was 12.2%, higher than both timothy hay (7.3%) and corn (1.2%). Normally, 80% of the ash contained in rice straw is silica [29], and silica limits the digestion of fiber because it prevents bacterial colonization [30]. Thus, the replacement of rice straw with timothy hay or corn gain significantly increased the growth and fermentation of rumen microbes.

When considering the associative effects, the observed NH_3_-N concentrations were significantly higher than the predicted concentrations in the RT, RC, and RTC treatments. An increase in the NH_3_-N concentration reflects greater catabolism of proteins and non-proteins [31,32], and may also indicate improvements in the rumen environment [33,34]. The growth of rumen microbes is stimulated by the digestibility of substrates that increases the NH_3_-N concentration [35]. Niderkorn et al. [36] tested the associative effects of legumes and grass and found that the observed NH_3_-N content was significantly higher than the predicted value in the legume-grass mixture. The mixture provided the rumen microorganisms with more nitrogen and other chemicals, and the interactions of these chemicals enhanced microbial growth and fermentation [36].

The observed TDMD values were significantly higher than the predicted values (*p* < 0.05; except RT), demonstrating a positive associative effect on dry matter digestibility; this result is consistent with other reports. Cho et al. [37] reported positive associative effects on the dry matter digestibility, organic matter digestibility, and organic matter effective degradability at an appropriate feeding ratio of corn grain and pasture forage. The authors attributed these results to improved ruminal digestion and the asynchrony of nutrient release by the rumen microorganisms, e.g., the release of soluble carbohydrate from grain seemed to improve the digestibility of the cell wall [37]. A corn grain-supplemented fescue hay diet also resulted in improved nutrient digestibility and rumen microorganism growth [38].

In the RTC treatment, the observed value of total VFA was significantly higher than the predicted value. This might be due to a higher NFC content from the corn grain and easily fermentable cellulose and hemicellulose from the timothy hay. Other work has shown that easily fermentable cellulose and hemicellulose increased the number of cellulolytic bacteria, stimulating the digestibility of other less degradable fiber sources in the diet [39]. Sun et al. [13] reported positive associative effects for the in vitro gas, total VFA, and microbial protein production when alfalfa hay was fermented with corn silage. Together, these results suggest that the combination of rice straw, timothy hay, and corn increases the NFCs from corn grain and the easily fermentable cellulose and hemicellulose from timothy hay, which influences the rumen microorganisms and fermentation environment and produces more fermentable substances to increase the total VFA.

There was a trend for the observed values to be higher than predicted values for *V_max_*, *k*_1_, and *k*_2_ in the RTC treatment. Within the same treatment, the observed *V*_1*max*_ was also significantly higher than the predicted value, and the discrete lag time observed was significantly lower than the predicted value. A higher *V*_1*max*_ and lower lag time indicate that the soluble fraction constitutes a substrate of rapid fermentation that facilitated the adhesion and colonization of microorganisms, thereby increasing fermentation and reducing the lag period [40]. Other studies have observed similar results, documenting a reduced lag time in the initiation of the in vitro digestion of fibers due to the positive associative effects of tropical grasses and legumes [41,42]. Measuring the in vitro gas production is a popular technique for determining the forage digestion characteristics and kinetics of fermentation. Rumen microorganisms ferment substrates into CO_2_, CH_4_, H_2_, and VFA [6], and the increase in gas production may be related to increased VFA, caused by the interaction of fiber and non-fiber decomposing microorganisms or ruminal bacteria and protozoa [6,7]. When the rice straw was fermented with timothy hay and corn grain, interactions among the rumen microorganisms increased substrate fermentation and produced more gas and VFA, which resulted in the higher *V*_1*max*_ and a shorter lag time.

Nutrient deficiencies in the roughage may be detrimental to the growth of rumen microorganisms (e.g., nitrogen or sulfur) or the ruminant itself (e.g., phosphorus), but positive associative effects were commonly observed when other feed materials contained the needed nutrients. Some studies have suggested that the supplementation of forage with rapidly fermentable carbohydrate sources (i.e., corn grain) would improve microbial growth and feed degradability [43], even in suboptimal nitrogen conditions, because of adaptations in the microbial populations [44,45]. Niderkorn et al. [46] studied the associative effects of temperate climate grass and legumes and found an influence on rumen protein degradation between the legume tannins and grass protein. Improvements to the rumen microflora were also demonstrated by the associative effects of cocksfoot (*Dactylis glomerata*) and sainfoin (*Onobrychis viciifolia*) [36]. Thus, when rice straw was fermented with timothy hay and corn, non-fiber carbohydrate from the corn grain, and easily fermented cellulose and hemicellulose from the timothy hay may have contributed to the positive associative effects.

## 5. Conclusions

This study demonstrated associative effects in ruminal fermentability (gas production kinetics and rumen parameters) when rice straw was co-fermented with timothy hay and corn grain. There was a trend of positive associative effect between rice straw and timothy hay in ruminal digestion; the increase in ruminal fermentability of rice straw was more significant when corn was supplemented. Our results suggest that the feed value of rice straw can be increased by the associative effects of different ingredients and that such factors should be considered in diet formulations.

## Figures and Tables

**Figure 1 animals-10-00325-f001:**
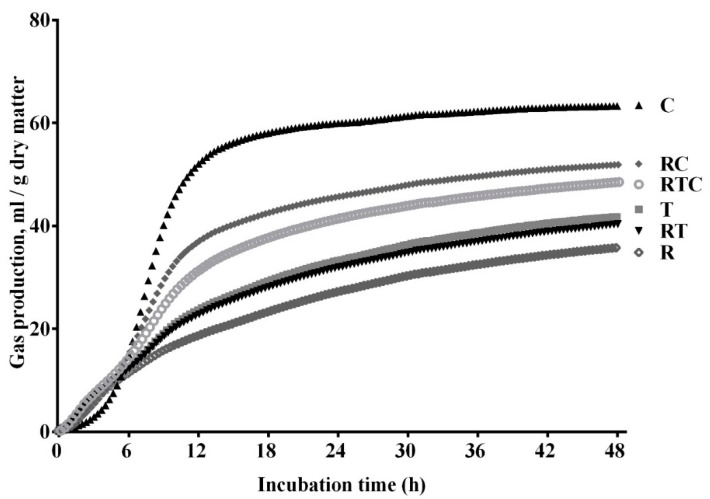
Gas production profiles of six treatments (R, 100% rice straw; T, 100% timothy hay; C, 100% corn grain; RT, 50% rice straw and 50% timothy hay; RC, 50% rice straw and 50% corn grain; RTC, 50% rice straw, 25% timothy hay, and 25% corn grain).

**Table 1 animals-10-00325-t001:** Chemical composition of rice straw, timothy hay, and corn grain (g/kg dry matter or as stated).

Items ^1^	Rice Straw	Timothy Hay	Corn Grain
DM, g/kg as fed	923	918	902
OM	877	926	988
CP	51	79	77
SOLP	16	26	8
NDICP	13	10	2
ADICP	12	8	1
aNDF	721	630	80
ADF	507	403	47
ADL	66	62	19
Ether extract	20	13	39
Ash	122	73	12
Ca	4.1	2.2	0.1
P	1.2	2.3	3.2
K	18.5	20.7	4.0
Na	1.4	0.3	0.1
Cl	6.8	8.0	0.6
S	1.6	1.3	0.9
Mg	1.7	1.5	1.2
TDN	491	555	878
NEm, MJ/kg DM	3.8	5.2	8.9
NEg, MJ/kg DM	1.5	2.9	6.2
Total carbohydrates	807	834	872
NFC	99	214	794

^1^ DM: dry matter, OM: organic matter, CP: crude protein, SOLP: soluble CP, NDICP: neutral detergent insoluble CP, ADICP: acid detergent insoluble CP, aNDF: neutral detergent fiber analyzed using a heat stable amylase and expressed inclusive of residual ash, ADF: acid detergent fiber, ADL: acid detergent lignin, Ca: calcium, P: phosphorus, K: potassium, Na: sodium, Cl: chlorine, S: sulfur, Mg: magnesium, TDN: total digestible nutrients, ME: metabolizable energy, NEm: net energy for maintenance, NEg: net energy for growth, NFC: non-fiber carbohydrate.

**Table 2 animals-10-00325-t002:** In vitro fermentation characteristics and kinetic parameters of gas production after 48 h incubation.

Items ^2^	Treatment ^1^						SEM	*p*-Value
	R	T	C	RT	RC	RTC		
pH	6.64 ^a^	6.51 ^c^	6.24 ^e^	6.55 ^b^	6.43 ^d^	6.50 ^c^	0.012	<0.001
NH_3_-N (mg/dL)	35.0 ^b^	39.1 ^a^	35.4 ^b^	38.2 ^a^	38.5 ^a^	38.6 ^a^	0.47	<0.001
Total VFA (mM)	70	78	94	75	83	80	8.2	0.148
Acetate (mmol/mol)	590	606	539	605	567	594	22.2	0.070
Propionate (mmol/mol)	231 ^b^	229 ^b^	266 ^a^	228 ^b^	236 ^a,b^	224 ^b^	10.0	0.014
Isobutyrate (mmol/mol)	20	19	17	19	19	18	0.7	0.112
Butyrate (mmol/mol)	91 ^b^	86 ^b^	125 ^a^	88 ^b^	118 ^a^	104 ^a^	11.7	0.024
Isovalerate (mmol/mol)	35	30	27	31	32	31	2.4	0.128
Valerate (mmol/mol)	33	30	26	30	29	29	2.5	0.343
A/P ratio	2.55	2.65	2.03	2.67	2.42	2.67	0.252	0.053
TDMD (%)	62 ^e^	74 ^c,d^	97 ^a^	71 ^d^	81 ^b^	77 ^b,c^	1.3	<0.001
NDFD (%)	47 ^b^	59 ^a^	62 ^a^	56 ^a,b^	54 ^a,b^	58 ^a,b^	2.4	<0.001
*V* _1*max*_	67 ^c^	93 ^c^	290 ^a^	87 ^c^	187 ^b^	160 ^b^	16.6	<0.001
*k* _1_	0.16	0.13	0.17	0.12	0.14	0.11	0.019	0.194
*V* _2*max*_	124	133	63	123	88	103	19.2	0.167
*k* _2_	0.03 ^b^	0.03 ^b^	0.04 ^a^	0.03 ^b^	0.03 ^b^	0.03 ^b^	0.002	0.001
*V_max_*	191 ^b^	226 ^b^	353 ^a^	210 ^b^	275 ^b^	262 ^b^	21.2	0.002
Lag	1.8 ^b^	2.8 ^b^	4.7 ^a^	2.2 ^b^	3.5 ^b^	2.4 ^b^	0.32	<0.001

^1^ R, 100% rice straw; T, 100% timothy hay; C, 100% corn grain; RT, 50% rice straw and 50% timothy hay; RC, 50% rice straw and 50% corn grain; RTC, 50% rice straw and 25% timothy hay and 25% corn grain. ^2^ VFA, total volatile fatty acids; TDMD, true dry matter digestibility; NDFD, neutral detergent fiber digestibility; *k*_1_, maximum rate of gas production (h^−1^) of the fast pool; *k*_2_, maximum rate of gas production (h^−1^) of the slow pool; *V*_1*max*_, maximum gas production of the fast pool (mL); *V*_2*max*_, maximum gas production of the slow pool (mL); lag, discrete lag time (h); *V_max_*, asymptotic total gas production (mL; *V*_1*max*_ + *V*_2*max*_). ^a–d^ Means within the same row with different superscripts differ within the treatments (*p* < 0.05).

**Table 3 animals-10-00325-t003:** Probability of significance of associative effects after 48 h fermentation.

Items ^2^	Treatment RT ^1^		SED	*p*-Value	Treatment RC ^1^		SED	*p*-Value	Treatment RTC ^1^		SED	*p*-Value
	RT (pre)	RT (obs)			RC (pre)	RC (obs)			RTC (pre)	RTC (obs)		
pH	6.57	6.55	0.006	0.074	6.44	6.43	0.003	0.184	6.50	6.51	0.003	0.423
NH_3_-N (mg/dL)	37.1	38.2	0.16	0.019	35.2	38.5	0.47	0.017	36.1	38.6	0.35	0.019
Total VFA (mM)	74	75	2.9	0.744	82	83	6.2	0.935	78	80	2.3	0.032
Acetate (mmol/mol)	598	605	5.6	0.329	564	567	20.4	0.912	581	594	12.1	0.403
Propionate (mmol/mol)	230	228	3.4	0.545	248	236	9.5	0.318	239	224	6.0	0.122
Isobutyrate (mmol/mol)	19	19	0.5	0.195	18	19	1.5	0.805	19	18	0.9	0.524
Butyrate (mmol/mol)	89	88	1.9	0.786	108	118	6.2	0.264	98	104	3.4	0.224
Isovalerate (mmol/mol)	33	31	0.6	0.092	31	32	2.4	0.697	32	31	1.1	0.589
Valerate (mmol/mol)	31	30	0.5	0.098	29	29	1.5	0.590	30	29	0.8	0.160
A/P ratio	2.62	2.67	0.07	0.412	2.35	2.42	0.18	0.541	2.48	2.67	0.14	0.241
TDMD (%)	68.2	71.4	1.21	0.120	79.6	81.2	0.41	0.042	73.9	77.7	1.40	0.046
NDFD (%)	52.7	56.4	1.24	0.097	54.3	53.7	1.93	0.797	53.5	57.7	3.18	0.321
*V* _1*ma*x_	80	87	6.3	0.370	178	187	21.5	0.714	129	160	7.7	0.023
*k* _1_	0.145	0.118	0.013	0.183	0.166	0.141	0.018	0.285	0.155	0.105	0.012	0.053
*V* _2*max*_	128	123	8.5	0.597	94	88	7.7	0.563	111	103	12.8	0.578
*k* _2_	0.028	0.027	0.001	0.054	0.033	0.032	0.001	0.222	0.030	0.028	0.001	0.055
*V_max_*	208	210	2.83	0.573	272	275	13.84	0.810	240	262	5.4	0.053
Lag	2.3	2.2	0.33	0.819	3.2	3.5	0.36	0.554	2.8	2.4	0.14	0.041

^1^ pre, predicted (the value of composited feeds with the weighted mean of individual feeds); obs, observed (experimental data); RT, 50% rice straw and 50% timothy hay; RC, 50% rice straw and 50% corn grain; RTC, 50% rice straw, 25% timothy hay, and 25% corn grain, SED; standard error of difference. ^2^ VFA, total volatile fatty acids; TDMD, true dry matter digestibility; NDFD, neutral detergent fiber digestibility; *k*_1_, maximum rate of gas production (h^−1^) of the fast pool; *k*_2_, maximum rate of gas production (h^−1^) of the slow pool; *V*_1*ma*x_, maximum gas production of the fast pool (mL); *V*_2*max*_, maximum gas production of the slow pool (mL); lag, discrete lag time (h); *V_max_*, asymptotic total gas production (mL; *V*_1*max*_ + *V*_2*max*_).

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
