# Peer review of "Evaluation of the Associative Effects of Rice Straw with Timothy Hay and Corn Grain Using an In Vitro Ruminal Gas Production Technique"

_animals, 2020, doi:10.3390/ani10020325_

Round 1

Reviewer 1 Report

Introduction: It is better to review the uses, effects and shortcomings of timothy in this party. The background and logic here is obviously weak.

Conclusion: the authors should give suggestions on the use of straw rice and timothy for practice in this part.

Author Response

Reviewers' Comments:

#1 reviewer:

Introduction: It is better to review the uses, effects and shortcomings of timothy in this party. The background and logic here is obviously weak.

[Response] The authors appreciate the reviewer’s reasonable comment. As suggested, we added some sentences in the introduction section as follows:

To date, however, no studies have tested the associative effects of rice straw with timothy hay and corn grain, which are commonly co-fed ruminants such as beef cattle. Especially, timothy hay is well known for its good quality and contributes a major proportion of imported forages in Korea [13]. Compared to rice straw, timothy hay contains easily digestible carbohydrates, so that it has higher digestibility of fiber and dry matter [14]. Timothy hay, however, is two to three times more expensive than rice straw, and thus it is mainly used as a forage source for calf and lactating dairy cows and as a supplement for rice straw for growing cattle in Korea [15].

Conclusion: the authors should give suggestions on the use of straw rice and timothy for practice in this part.

[Response] The authors appreciate the reviewer’s reasonable comment. As suggested, we revised the conclusion section as follows:

This study demonstrated associative effects in ruminal fermentability (gas production kinetics and rumen parameters) when rice straw was co-fermented with timothy hay and corn grain. There was a trend of positive associative effect between rice straw and timothy in ruminal digestion; the increase in ruminal fermentability of rice straw was more significant when corn was supplemented. Our results suggest that the feed value of rice straw can be increased by the associative effects of different ingredients and that such factors should be considered in diet formulations.

Reviewer 2 Report

The manuscript has been improved following the indications given and, as far as I am concerned, can now be published.

Author Response

The authors appreciate the encouraging comment by the reviewer.

Reviewer 3 Report

I'm happy with the revision and have no further comments - go ahead.

Author Response

The authors appreciate the encouraging comment by the reviewer.

This manuscript is a resubmission of an earlier submission. The following is a list of the peer review reports and author responses from that submission.

Round 1

Reviewer 1 Report

I suggest the authors to add some data like NDF digestibility, ADF digestibility and microbial crude protein (MCP) as well, as those substrate used in the current work are all roughage which is to be used in ruminants animal and the fiber characteristics is important for rumen fermentation and motility.

Author Response

#1 reviewer:

I suggest the authors to add some data like NDF digestibility, ADF digestibility and microbial crude protein (MCP) as well, as those substrate used in the current work are all roughage which is to be used in ruminants animal and the fiber characteristics is important for rumen fermentation and motility. 

[Response] The authors appreciate the reviewer’s constructive comment. The manuscript presents the true DM digestibility that is considered the most important and should be focused on assessing the associative effects. Nevertheless, we agree with the reviewer that showing NDF digestibility will help readers assess changes in fiber digestion. Therefore, we added NDF digestibility in Table 2 as the reviewer suggested. The changes in NDF digestibility among treatments should be correlated with those in ADF digestibility, and the former better represents digestion of total fiber. Microbial crude protein (MCP) production after in vitro ruminal fermentation was not measured in this study; however, an increase in NH3-N concentration indirectly indicated an increase in MCP in this experimental setting. We will consider including measurement of MCP in our future study as the reviewer suggested.

Reviewer 2 Report

I suggest authors to concentrate, in the discussion, on the most important statistically significant differences between treatments, both for the six treatments (table 2 and figure 1) and for the comparison between predicted and observed values (table 3). it would be interesting to compare the magnitude of the associative effects found in the present study with that of other experiments published, to see if there is consistency or not.

For specific notes see the attached file.

Author Response

#2 reviewer:

I suggest authors to concentrate, in the discussion, on the most important statistically significant differences between treatments, both for the six treatments (table 2 and figure 1) and for the comparison between predicted and observed values (table 3). it would be interesting to compare the magnitude of the associative effects found in the present study  

[Response] The authors appreciate the reviewer’s constructive comments. After carefully addressed all the scientific and instructive comments of the reviewer, we believe that the quality of the manuscript has greatly improved.

Line 1 - Replace “Corn” with “Corn Grain”

[Response] As suggested, we replaced corn with ‘corn grain’ throughout the manuscript.

Line 31 – Replace “(pH, NH3-N, and volatile fatty acids [VFA], true dry” with “pH, NH3-N, volatile fatty acids [VFA], and true dry”

[Response] We apologize for the typo. Changed as suggested.

Line 68 – “few studies”. Are you sure it is not “no studies”? Otherwise cite references.

[Response] As far as we know there is no scientific publications showed associative effects of rice straw with timothy hay and corn. Changed ‘few’ to ‘no’ as suggested.

Lines 132-134 – The diet of donor animals seems unusual, with a high energy and a fairly high protein content. Why not using a standard diet for dry cows based on hay or grass silage? The analysis of both corn silage and concentrate are given on a DM basis? For corn silage the content of 921 g DM/kg is clearly wrong and the content of 121 g CP is very high, even if referred to DM. For the fibrt content I suggest to use NDF and ADF instead of CF. I know feed compounders normally use crude fibre, but knowing the composition of the feedstuff it is possible to compute the detergent fibre fractions.

[Response] The authors apologize for the typo. Replaced “921 g DM/kg” to “232 g DM/kg” as suggested. We totally agree with the reviewer that NDF is better value for fiber than crude fiber. After re-analyzing the samples, NDF and ADF are reported in the revision. We agree with the reviewer that 121 g/kg CP is higher than usual values; however, we obtained the same value with the re-analysis. And, crop silages (corn, barley, rice straw), hay (mostly straw), or both with a concentrate mix is the typical diet for beef cattle and growing or dry dairy cattle in Korea.

Lines 210-212 - Replace “The total VFA and TDMD were…” with “The TDMD was…”.

[Response] Replaced as suggested.

Line 212 – After “…for rice straw” and before “Moreover…” add the sentence: “The same trend, although not significant, was observed for the total VFA production.”

[Response] As suggested, the sentence The same trend, although not significant, was observed for the total VFA production.” was added.

Line 213 – “(P <0.01)”. In table 2 the stated level of significance is P < 0.05.

[Response] The authors apologize for the typo and it was revised as suggested.

Line 214 – Replace “…groups” with “group”.

[Response] the words of ‘groups’ were deleted throughout the manuscript.

Lines 213, 215, 217, 219. I suggest to delete “(RT, RC, and RTC)”, since redundant.

[Response] Deleted as suggested.

Line 237-238 – Replace “Compared with the predicted and observed data” with “Comparing the observed data with the predicted ones”.

[Response] Replaced as suggested.

Lines 243-245 – Replace “RTC treatment significantly decreased the discrete lag time (P < 0.05) and significantly increased the observed value for Vmax1 (P < 0.05) t compared with the predicted value.” With “RTC treatment significantly decreased the observed value for the discrete lag time (P < 0.05) and significantly increased the observed value for Vmax1 (P < 0.05) compared with the predicted values.”

[Response] Replaced as suggested.

Lines 279-281 – The comments are not consistent with the data reported in table 3. Also the language has to be improved in this sentence. Looking at data of table 3 for the VFA it seems that the sentence should be something like: “… the observed value was significantly higher than the predicted one in the RTC treatment”. I would skip to mention a numerical difference if the P value is far from that chosen to highlight a trend (P < 0.10) (see line 198), as is the case for total VFA production differences between observed and predicted values in RT and RC treatments.

[Response] The authors appreciate the reviewer’s constructive comments. We revised the whole paragraph. Also, we carefully revised the text throughout the manuscript to avoid stating a numerical difference when the P value is higher than 0.1.

Lines 293-294 – See the comment above. Underline just the trend for a higher observed AsyGP in comparison with the predicted value for RTC treatment.

[Response] Revised as suggested. Changed sentence with “There was an increased trend of observed value than predicted one in Vmax of RTC treatment.

Lines 296-299 – So what? I do not understand the comment and the relevance of the cited study with your results.

[Response] The authors appreciate the reviewer’s comment. We revised the paragraph as suggested.

Line 301 – As previously, specify that the higher observed TDMD as compared to the predicted one is referred only to the RC and RTC treatments, not to RT.

[Response] The authors apologize for the typo. added “except (RT)” as suggested.

Line 316 – Replace “Low” with “low”.

[Response] replaced as suggested.

Line 320 – Replace “…will lead to associative an effect” with “…will lead to an associative effect”

[Response] The authors apologize for the typo and replaced as suggested.

Lines 351-354 – This sentence is more a general assessment on the importance of the associative effects, than a real discussion of what obtained in the experiment and compared to what found in literature. Moreover: why three nutrients, and which ones?

[Response] The authors appreciate the reviewer’s constructive comment. We revised the entire discussion session to discuss more of our specific results as suggested.

Table 1 – In the title replace “(% of dry matter)” with “(all data, except DM, are expressed on dry matter)”.

[Response] Revised to “g/kg dry matter or as stated”

In columns 3 and 4 I would replace “timothy” with “timothy hay” and “Corn” with “Corn grain”. NFC: the value reported for rice straw (131.80 g/kg DM) seems too high; computing it as (OM-CP-EE-aNDF) it results to be 85.70. For the other two treatments the gap is smaller (204,20 vs 215.80, and 792.4 vs 794.00).

[Response] The authors appreciate the reviewer’s comment. We apologize there was an error in calculating the NFC value of rice straw. However, the other values are correct since we estimate NFC as OM – CP – EE – (aNDF – NDICP).

Table 2 – k1 and K2: why capital letter for the second? Lag: why values increase from R (1.8 h) to T (2.8 h), and to C (4.7 h)? Should not be the other way round, with a decrease of the lag phase passing from a fibrous feed as R to a starchy feed as corn grain? The same question is valid for the lag values reported for the other three treatments.

[Response] The authors apologize for the typo and revised “K” to “k” throughout the manuscript. Regarding the lags, we do not fully understand why this happen; however, supplementation of highly fermentable carbohydrates does not always decrease the discrete lag time of fermentation. For example, Mertens and Loften (1980; J. Dairy Sci. 63:1437-1446) observed longer lag times when coastal bermudagrass and fescue were supplemented with starch. They suggested this might be due to the sensitive of rumen cellulolytic bacteria to the acidity condition in the initial time of incubation.

Table 3 – Again K in capital letters: is that right?

[Response] The authors apologize for the typo and revised “K” to “k” throughout the manuscript.

Reviewer 3 Report

this is a rather descriptive study. No causalities or whatever insights are given. The discussion is repetitive to the results. The anticipated effects are commonly described but in no case explained. 
it is a poor outcome to state in the last paragraph ( end of discussion) that further research is necessary ( this is always the case): 

A thorough study of associative effects at the metabolic level can provide a
351 more comprehensive understanding of the underlying mechanism. In order to systematically
352 understand the mechanism of associative effects, it is necessary to further study the fiber and
353 protein in feed, associative effects of fiber and fat, easily degradable fiber and non-degradable
354 fiber, and three or even a variety of nutrients. Rumen microbes may help to explain the
355 associative effects between forage and grain; thus, further rumen microbiological studies are
356 recommended.

Author Response

#3 reviewer:

This is a rather descriptive study. No causalities or whatever insights are given. The discussion is repetitive to the results. The anticipated effects are commonly described but in no case explained. It is a poor outcome to state in the last paragraph (end of discussion) that further research is necessary (this is always the case): 

A thorough study of associative effects at the metabolic level can provide a
351 more comprehensive understanding of the underlying mechanism. In order to systematically
352 understand the mechanism of associative effects, it is necessary to further study the fiber and
353 protein in feed, associative effects of fiber and fat, easily degradable fiber and non-degradable
354 fiber, and three or even a variety of nutrients. Rumen microbes may help to explain the
355 associative effects between forage and grain; thus, further rumen microbiological studies are
356 recommended.

[Response] The authors appreciate the reviewer’s comment. We agreed with reviewer’s comment and revised the entire discussion section. We have added more explanations and discussions that are particularly relevant to the findings from this study. We also deleted the last paragraph as suggested.
